# Water striders adjust leg movement speed to optimize takeoff velocity for their morphology

Eunjin Yang[1], Jae Hak Son[2], Sang-im Lee[3], Piotr G. Jablonski[2,4] & Ho-Young Kim[1,5]

Water striders are water-walking insects that can jump upwards from the water surface. Quick jumps allow striders to avoid sudden dangers such as predators' attacks, and therefore their jumping is expected to be shaped by natural selection for optimal performance. Related species with different morphological constraints could require different jumping mechanics to successfully avoid predation. Here we show that jumping striders tune their leg rotation speed to reach the maximum jumping speed that water surface allows. We find that the leg stroke speeds of water strider species with different leg morphologies correspond to mathematically calculated morphology-specific optima that maximize vertical takeoff velocity by fully exploiting the capillary force of water. These results improve the understanding of correlated evolution between morphology and leg movements in small jumping insects, and provide a theoretical basis to develop biomimetic technology in semi-aquatic environments.

[1] Department of Mechanical and Aerospace Engineering, Seoul National University, Seoul 08826, Korea. [2] Laboratory of Behavioral Ecology and Evolution, School of Biological Sciences, Seoul National University, Seoul 08826, Korea. [3] Daegu-Gyeongbuk Institute of Science and Technology School of Undergraduate Studies, Daegu 42988, Korea. [4] Museum and Institute of Zoology, Polish Academy of Sciences, Wilcza 64, Warszawa 00-679, Poland. [5] Big Data Institute, Seoul National University, Seoul 08826, Korea. Correspondence and requests for materials should be addressed to P.G.J. (email: piotrjab@behecolpiotrsangim.org) or to H.-Y.K. (email: hyk@snu.ac.kr).

It is widely known that the superhydrophobic hairy legs of water striders enable them to float only on tarsi[1,2], and that the striders transport the momentum via vortices and capillary waves to propel themselves across the water surface[1,3–5]. But what are the mechanical characteristics of jumping off the water surface? Vertical and near-vertical jumps are performed in natural habitats in a series of frequent jumps that are triggered by attacks of predators, such as fish and backswimmers (predatory insects of the genus *Notonecta*), from under the water surface[6,7]. While it was suggested that water striders push the water surface with their legs to generate the upward capillary force on jumping[3,8–10], the mechanics of this swift mode of locomotion have not been fully understood. Recently, a robotic water strider was built that uses the basic principle of momentum transfer observed in one of the larger species of water striders, *Aquarius paludum*[11]. A simple mechanical model of interactions between *A. paludum* legs and the water surface has been created to aid designing the robot. But it is still uncertain whether the combinations of species-specific morphology and leg movements observed in water striders maximize the insects' jumping performance as expected from natural selection for predation avoidance. Upon the basis of the initial kinematic calculations[11] and the insights from high speed videos of jumping, we develop here a theoretical model of insect leg movements that enables the predictions of conditions for optimal jump performance in water striders. We then empirically verify the optimal predictions using individuals from five water strider species of different body sizes and leg morphologies. We show that, despite having different morphological constraints on leg dimensions, species tune their leg rotation speed to optimize the takeoff velocity from the water surface.

## Results

**General description of a water strider's jumping on water.** From high-speed videos (see Methods for details) of three species of water striders (*Gerris latiabdominis*, *G. gracilicornis* and *A. paludum*), we observed near vertical jumps (with trajectories steeper than 60° to the horizontal; Supplementary Video 1) in order to develop the mathematical model of the vertical component of jumping. At rest, a water strider (Fig. 1a) supports its weight using all of six legs with its body centre located at $y_i$ (see list of symbols and their descriptions in Table 1) above the water surface. Once the insect initiates a jump by pushing the surface with the middle and hind legs downwards, dimples are made (Fig. 1a), which enables the insect to control its direction and speed of the jump by transferring momentum to the water. The dimple depth $h$ (Fig. 1b), measured from the unperturbed water surface, increases and then decreases with time $t$ (Fig. 1f). The dimple depth reaches its maximum $h_m$ at the time $t_m$, which divides the jump into two stages: the pushing ($t < t_m$) and the closing ($t > t_m$) stage. The average downward velocity of four legs with respect to horizontal plane through insect's body centre $v_s$ is bell-shaped over time (Fig. 1g), and the upward velocity of the insect body $v$ overtakes $v_s$ at the moment of reaching the maximum dimple depth, $t = t_m$, when the vertical growth rate of the dimple ($v_s - v$) becomes zero and its corresponding maximal depth is denoted as $h_m$. Hence, in the pushing stage the downward velocity of legs is larger than the upward velocity of the insect body ($v_s > v$), but in the closing stage $v_s < v$.

In the pushing stage, the tarsus and tibia of each leg (Fig. 1c) remain in contact with the water surface until $t_m$ ($t_m \approx 13$ ms in Fig. 1e). Thus, the average wetted length of the legs $l_w$ is assumed to be almost constant and equal to the average of sum of tarsi and tibiae lengths of four legs $l_t$. In the closing stage ($t > t_m$), the legs continue to come close together and slide on the water surface

towards the body while gradually disengaging themselves from the water surface causing a decrease in $l_w$. Owing to the decreasing wetted length $l_w$ that interacts with the water surface, the increase rate in upward velocity of the insect body in the closing stage is lower than that in the pushing stage as shown in Fig. 1g.

Based on these observations, we build a model to calculate the vertical force that produces the vertical jump as a function of the depth of the dimple formed by the legs. The force can be used to estimate the takeoff velocity, which allows us to seek the optimal stroke condition that the water strider should perform to maximize jumping speed. Fast jumping is important for the water striders to escape from the predators, such as backswimmers or fish, attacking from under the water surface. Therefore, evolution of abilities to maximize takeoff speed is expected under natural selection.

**Theoretical model.** When a water strider strokes the water surface, forces of various origins are exerted on the insect's legs in addition to the capillary force ($F_s \sim \sigma l_w$), including pressure force ($F_p \sim \rho U^2 r l_w$), buoyancy ($F_b \sim \rho g r h l_w$), inertial force due to added mass ($F_a \sim \rho r^2 l_w U^2 / h$), viscous force ($F_v \sim \mu r l_w U / l_c$) and the weight of the water strider ($F_w \sim mg$). Here $\sigma$ is surface tension coefficient, $\rho$ is density of liquid, $U$ rate of vertical growth of dimple, $r$ radius of the leg, $g$ gravitational acceleration, $\mu$ viscosity of liquid and $l_c = [\sigma/(\rho g)]^{1/2}$ capillary length of water. Our theoretical analysis used the standard values, such as density of water, $\rho = 998$ kg m$^{-3}$; surface tension coefficient of water, $\sigma = 0.072$ N m$^{-1}$; viscosity of water, $\mu = 10^{-3}$ Pa·s; gravitational acceleration, $g = 9.8$ m s$^{-2}$ and average values of experimentally measured parameters for jumping *G. gracilicornis*, the medium-sized water strider species: wetted length, $l_w = 7.5$ mm (evaluated from the average length of four pairs of tibia and tarsus); leg radius, $r = 50$ μm (measured at the middle of the tibia); body mass of water strider, $m = 30$ mg; representative leg descending speed, $U = 0.15$ m s$^{-1}$; and depth of dimple, $h = 3$ mm. The ratios of the other forces to the capillary force are scaled as $F_p/F_s \sim 10^{-2}$, $F_b/F_s \sim 10^{-2}$, $F_a/F_s \sim 10^{-4}$, $F_v/F_s \sim 10^{-5}$ and $F_w/F_s \sim 10^{-1}$. These ratios suggest that the capillary force dominates over the other forces. Moreover, the ratio of energy loss $E_d$ when the leg becomes detached from the water surface to kinetic energy of the water strider taking off the surface, $E_k \sim mv^2$, are scaled as $E_d/E_k \sim 10^{-4}$, implying negligible energy loss due to wet adhesion. Here, the reported value of energy loss, $E_d$, of a leg of a water strider via detachment from the water surface is of the order of $10^{-9}$ J[12], and $v \sim 1$ m s$^{-1}$.

Hence, the seemingly complex phenomenon of jumping on the water surface can be simplified as a surface tension-dominant interaction of a long thin flexible cylinder with the water surface. Assuming that all the legs involved in the propulsion move synchronously and leave the surface at the same time (Supplementary Fig. 1 and Supplementary Note 1), the upward force $F$ on four legs, which is equivalent to the weight of water displaced by the legs (creating the dimples), is estimated by modifying Vella's model[13] (Supplementary Note 2)

$$F = 8\rho g l_c C l_w h \left\{ 1 - [h/(2l_c)]^2 \right\}^{1/2} \qquad (1)$$

where $C$ is the flexibility factor depending on the scaled leg length $L_f = l_w/l_{ec}$, and $l_{ec} = (Bl_c/\sigma)^{1/4}$ being the modified elastocapillary length of the leg with the bending rigidity $B = \pi E r^4/4$, $E$ being Young's modulus of insect cuticle and $r$ being radius of the leg. We approximated $C$ as $C \approx (1 + 0.082 L_f^{3.3})^{-1}$ for $L_f < 2$ and $C l_w$ could indicate the effective wetted leg length (Supplementary Fig. 2 and Supplementary Note 2). The force $F$ increases

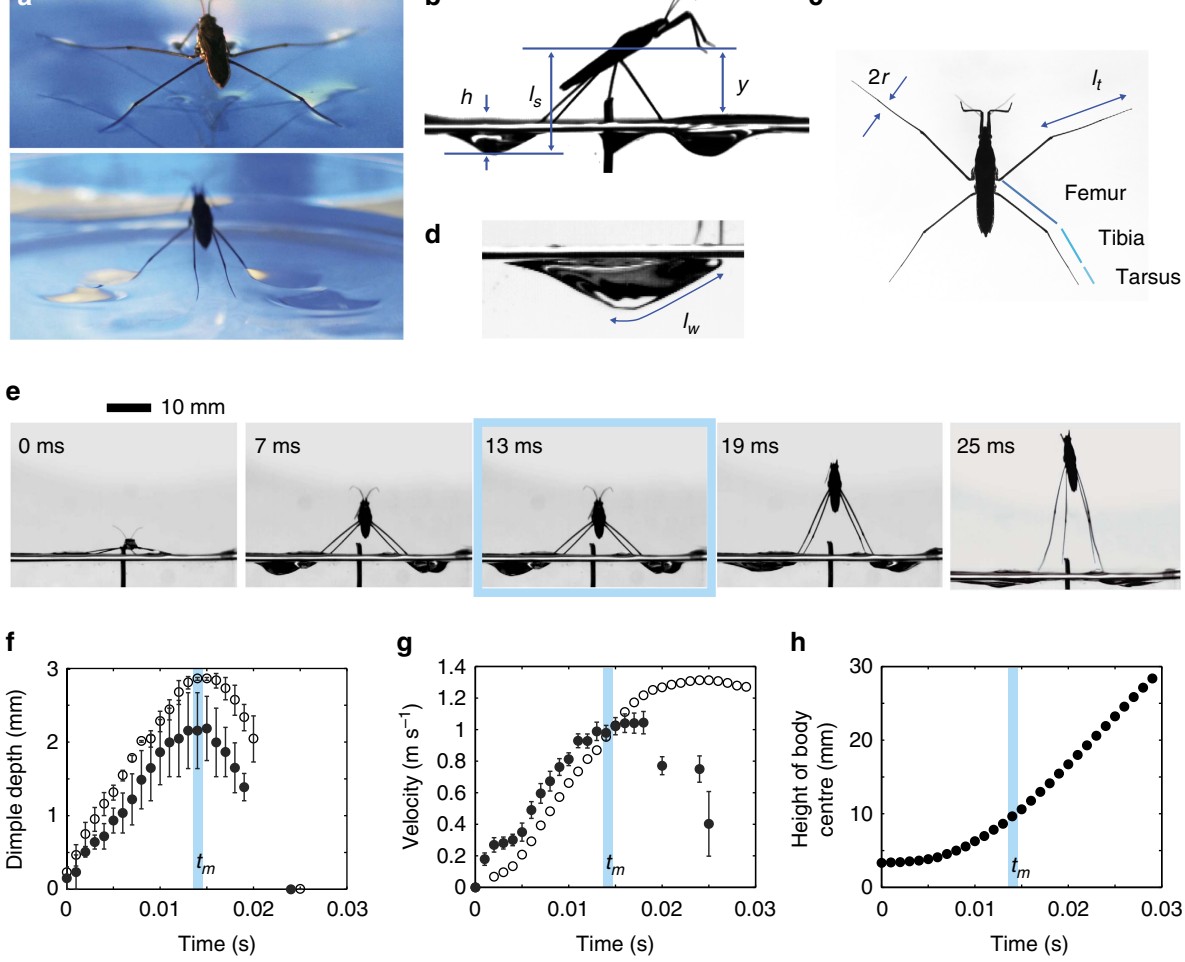

**Figure 1 | Jumping of a water strider.** (**a**) A water strider (male *Aquarius paludum* with a body mass of 37.2 mg and an average length of middle and hind legs of 22.1 mm) that rests and jumps on water. (**b-d**) Definitions of various lengths considered in this study. (**b**) The vertical lengths including body centre location $y$, vertical distance from the tip of the legs to the horizontal plane through body centre $l_s$ and dimple depth $h$. (**c**) The lengths of legs including the radius $r$, and the length of tibia plus tarsus $l_t$. (**d**) The wetted length of the leg $l_w$. (**e**) A representative sequence of the jump of the water strider on the water surface. (**f-h**) Measurement data extracted from a movie corresponding to **e**. (**f**) Average dimple depth formed by the right and left legs (open circles for middle legs, filled circles for hind legs) during the jump. Error bars indicate standard deviation between right and left legs. (**g**) Vertical velocity of the body centre $v$ (open circles) and the average downward velocity of the four legs with respect to the horizontal plane through the body centre $v_s$ (filled circles). Error bars indicate standard deviation among four legs. (**h**) Height of the body centre of the water strider during the jump. The vertical blue bar in **f-h** indicates the moment when the dimple reaches the maximal depth (the panel corresponding to 13 ms in **e** shows the dimple that reached maximal depth). Body velocity profile in **g** is the same data as that of water strider 2 in Koh *et al.*[11]

monotonically with the depth of dimple $h$ until the surface penetration occurs.

From momentum conservation, the upward velocity of the centre of mass of the insect can be determined as $v = \int F dt / m$, with $m$ being the insect's body mass. Then, the temporal change of the dimple depth $h(t)$ is given by

$$\frac{dh}{dt} = v_s - \frac{1}{m}\int F dt. \qquad (2)$$

We approximated the relationship between the downward linear velocity of the legs $v_s$ and the angular velocity of leg rotation $\omega$ as a sinusoidal function: $v_s = \omega \Delta l \sin(2\omega t)$, based on the vertical distance of the legs $l_s$, which was measured between the tip of the legs and the horizontal plane through the body centre and modelled as $l_s = \Delta l[1 - 1/2\cos(2\omega t)] + y_i$ during the stroke. Here the angular velocity of the leg rotation $\omega$ is assumed to not change during a jump. The angular velocity of leg rotation $\omega$ and the maximal (theoretical) downward reach of the legs

$\Delta l = l_l - y_i$, where $l_l$ is the average length of the four legs, are the parameters reflecting behavioural and morphological traits of each insect, respectively. This is one of the simplest models satisfying necessary conditions to imitate water striders' leg movements with respect to the body centre during the jump: the vertical distance of the legs $l_s$ increases from $y_i$ to $l_l$, while the downward linear velocity of the legs $v_s$ increases from zero and then decreases to zero with its maximum in the middle. We confirmed that this theoretical model matches well the movements of the real water striders' legs (Fig. 2a,b).

In the model, the angular velocity of leg rotation $\omega$ does not change during a jump. This concept is unrealistic, but to illustrate how close our theoretical approximation of leg rotation $\omega$ is to the real angular leg movements by jumping insects, we compared the time derivative of average angle of four legs with respect to the horizontal plane through the body centre, $\dot{\theta}$ (extracted from the video), with the value of $\omega$ that leads to a good match between empirical and modelled vertical distance of the legs $l_s$ (Fig. 2a)

**Table 1 | Explanations of the symbols in the model.**

| | |
|---|---|
| $\rho$ | Density of water |
| $g$ | Gravitational acceleration |
| $\sigma$ | Surface tension coefficient of water |
| $l_c = (\sigma/\rho g)^{1/2}$ | Capillary length |
| $m$ | Insect body mass |
| $l_l$ | Average leg length (femur + tibia + tarsus) of four legs of an individual |
| $l_t$ | Average length of the part of a leg of an individual that supports the insect on the surface during jump; in typical water striders this corresponds to the tibia plus tarsus length (average from the four legs of an individual: two midlegs and two hindlegs) |
| $l_w$ | Average wetted length of legs: length of tibia and tarsus ($l_t$) in the first 'pushing' stage of jump; in the second 'closing' stage of jump, wetted leg length gradually decreases |
| $E$ | Young's modulus of insect cuticle |
| $r$ | Species-specific average radius of four legs (tibia) |
| $B = \pi E r^4 / 4$ | Bending rigidity of a leg |
| $l_{ec} = (Bl_c/\sigma)^{1/4}$ | Modified elastocapillary length of a leg |
| $L_f = l_w/l_{ec}$ | Scaled leg length; function of wetted length of a leg, $l_w$, and its bending rigidity $B$ |
| $C$ | Flexibility factor; function of wetted length of a leg, $l_w$, and its bending rigidity $B$ |
| $Cl_w$ | Effective wetted leg length |
| $F$ | Total upward force on legs |
| $t$ | Time |
| $t_m$ | The moment when dimple reaches the maximal depth |
| $t_b$ | The instant of meniscus breaking |
| $t_c$ | The instant of the end of closing of the legs |
| $t_t$ | The instant of takeoff; the tips of escaping legs reach the zero depth position |
| $h$ | Dimple depth; average distance from the unperturbed water surface to the deepest point of the water dimples beneath four legs |
| $h_m$ | The maximal dimple depth reached during the jump |
| $y$ | Body centre location on vertical coordinate axis |
| $y_i$ | Initial body centre location on vertical coordinate axis; this represents the distance from body centre (located between leg bases) to the undisturbed water surface in the resting position of the water strider |
| $\Delta l = l_l - y_i$ | Maximal reach of the leg; the maximal distance the legs can reach from body centre |
| $l_s = h + y$ | Vertical distance from the tip of the legs to the horizontal plane through body centre, which changes during the stroke |
| $\theta$ | Average angle of femur with respect to the horizontal plane through body centre in a rotation plane of four legs |
| $\omega$ | Angular velocity of leg rotation of a jump |
| $v_s = \omega\Delta l \sin 2\omega t$ | Average downward velocity of the four legs with respect to the horizontal plane through body centre which changes during the stroke; function of the angular velocity of leg rotation $\omega$, maximal reach $\Delta l$ and time $t$ |
| $v$ | Vertical velocity of body centre |
| $v_t$ | Vertical component of takeoff velocity of body centre |
| $L = \Delta l/l_c$ | Dimensionless maximum downward reach of leg; the maximal distance the legs can reach downward from body centre expressed in the units of water capillary length |
| $H = h/l_c$ | Dimensionless dimple depth; dimple depth in units of water capillary length |
| $H_m = h_m/l_c$ | The maximal dimensionless dimple depth; maximal dimple depth expressed in units of capillary length |
| $\omega t$ | Phase of leg rotation; ranges from 0 to $\pi/2$, see Fig. 2 |
| $\Omega = \omega(l_c/g)^{1/2}$ | Dimensionless angular velocity of leg rotation |
| $M = m/(\rho l_c^2 C l_t)$ $\approx 8 Ba$ [Baudoin number $Ba = mg/(\sigma P)$ $P$ : perimeter of wetted length] | Dimensionless index of insect body mass; body mass with respect to the possible maximum mass of water that can be displaced by the leg. $M$ is a function of body mass and morphology represented by the total tibia plus tarsus length; as body mass increases and/or the length of tibia plus tarsus decreases, the $M$ value increases. The mass of water displaced is equivalent to the upward force from water surface |
| $V = v/(gl_c)^{1/2}$ | Dimensionless vertical velocity of insect body centre |
| $V_t = v_t/(gl_c)^{1/2}$ | Dimensionless vertical takeoff velocity of insect body centre |

and downward linear velocity of the legs $v_s$ (Fig. 2b). We calculated the value of $\omega$ for each jump ($\omega = v_{s,\max}/\Delta l$) with empirically measured $v_{s,\max}$ (maximum value of $v_s$) and $\Delta l$. We also calculated the time derivative of the empirically measured (from video) average angle of legs with respect to the horizontal plane through the body centre (Supplementary Fig. 3 and Supplementary Note 3). The value of $\dot\theta$ was not constant during the jump, but rather tended to quickly increase during the initial 8–10 ms of a jump, and then it fluctuated randomly (partly due to measurement error) indicating the apparent plateau as shown in Fig. 2c. Interestingly, the angle $\theta$ varies over time in a manner resulting in the changes of the vertical distance from the tip of the legs to the horizontal plane through body centre, $l_s$, approximating a sinusoidal function of $\omega t$. Note that

$l_s = \Delta l \sin\theta(t) \approx \Delta l[1 - \frac{1}{2}\cos(2\omega t)] + y_i$. Furthermore, the value of $\omega$ is similar to the time average of empirical value of the average angular velocity of the leg rotation $\dot\theta$ as seen in Fig. 2c. Therefore, we believe that our theoretical approach by using $\omega$ is a reasonable theoretical representation of the average angular velocity of leg rotation of vertically jumping water striders.

Combining equations (1) and (2), with the sinusoidal model of $v_s$, leads to a simple differential equation of the scaled dimple depth $H(\omega t)$ as

$$\frac{d^2 H}{d(\omega t)^2} + \frac{8}{\Omega^2 M}H\left(1 - H^2/4\right)^{1/2} - 2L\cos(2\omega t) = 0, \qquad (3)$$

where $H = h/l_c$ (dimensionless dimple depth), $\Omega = \omega(l_c/g)^{1/2}$

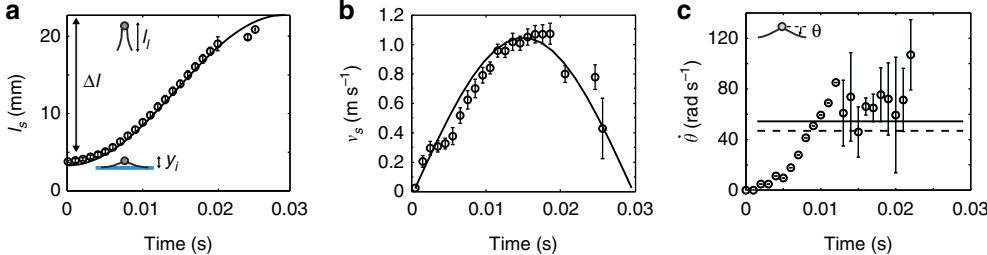

**Figure 2 | Comparison of empirical and modelled leg movements.** The solid lines correspond to the model assuming the sinusoidal model of leg rotation and the circles correspond to the average of measurement of four legs from the same movie used in Fig. 1e–h. The error bars indicate standard deviation among the four legs. (**a**) The average vertical distance between the body centre and distal end of legs ($l_s$) across the leg rotation cycle. (**b**) The average downward velocity of four legs with respect to body centre ($v_s$). (**c**) The average time derivative of the angle of legs with respect to the horizontal plane ($\dot{\theta}$). The dashed line indicates the time average of the measured values of $\dot{\theta}$ through the whole cycle and the solid line refers to the corresponding angular speed of leg rotation $\omega$ used in the model calculations of $l_s$ and $v_s$ in **a,b**.

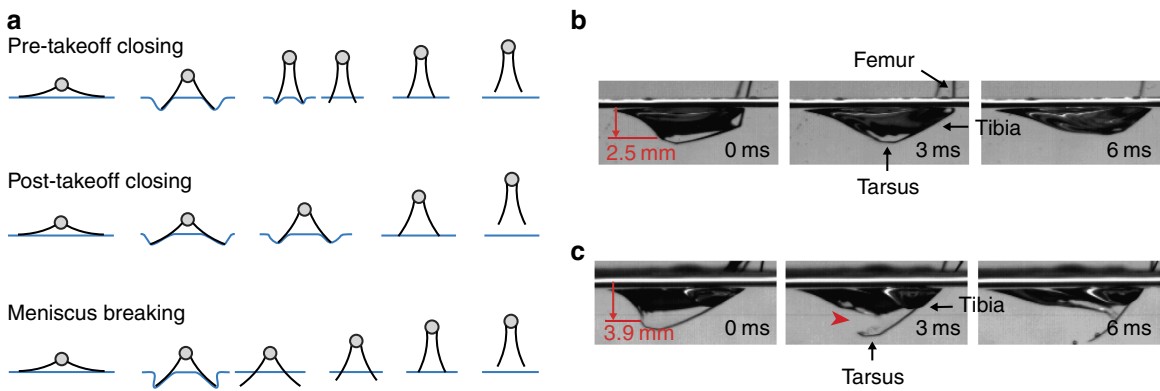

**Figure 3 | Different jump modes.** (**a**) Schematic representation of three modes of jump: pre-takeoff closing, post-takeoff closing and meniscus breaking jumps. (**b,c**) Enlarged images of the leg and dimple in a post-takeoff closing jump ($h_m = 2.5$ mm) and a meniscus breaking jump ($h_m > 3.9$ mm): (**b**) the leg that does not reach the sinking depth leaves the surface unpenetrated, (**c**) the leg pierces the surface just below the sinking depth. The red arrow at 3 ms indicates the rupture point of the water surface. The pre-takeoff closing jump was not observed in the experiments.

(dimensionless angular velocity of leg rotation), $M = m/(\rho l_c^2 C l_t)$ (dimensionless index of insect body mass with respect to the maximum mass of water that can be displaced by the leg of the total tibia plus tarsus length $l_t$, which is directly related to the maximum supporting force of the water surface), and $L = \Delta l/l_c$ (dimensionless maximum downward reach of leg). $M \approx 8 Ba$ where Baudoin number $Ba = mg/(\sigma P) = 1$, with $P$ the perimeter of wetted parts of legs, implies the maximum body weight that capillary force can support. Here, $\omega t$ is the phase of leg rotation, being 0 at the beginning and $\pi/2$ at the end of the stroke (the range $[0–\pi/2]$ is due to the sinusoidal approximation of the leg movements).

In the pushing stage ($v_s > v$), the observed visible wetted legs $l_w$ comprised tibia and tarsi, $l_t$. Therefore, the wetted length in the pushing stage is estimated to be $l_w \approx l_t$, leading to $M \approx m / (\rho l_c^2 C l_t)$. However, in the closing stage ($v_s < v$), the wetted leg length $l_w$ decreases gradually while the legs disengage themselves from the water (Supplementary Video 1). In addition, as the legs close in the closing stage their inclination angles from the water surface increase leading to an apparent decrease of flexibility factor $C$[13].

To consider these changes in $l_w$ and $C$, while solving the differential equation (3) of dimple growth and decay, we simplified the wetted length in the closing stage to $l_w = l_t(\pi/2 - \omega t)/(\pi/2 - \omega t_m)(H/H_m)$. This equation consists of three terms: the wetted length in pushing stage $l_t$, and the two terms that decrease with the increase of $\omega t$ and the corresponding decrease of $H$ respectively. Then, the temporal

change of the dimple depth can be obtained by solving the differential equation with two pairs of initial conditions: $H(0) \approx 0$ corresponding to negligible initial dimple depth formed by the weight of the insect, and $H'(0) = 0$ at the beginning of the pushing stage; $H(\omega t_m) = H_m$ and $H'(\omega t_m) = 0$ at the instant of maximum dimple depth when the closing stage starts. The takeoff velocity of the water strider $v_t$ is defined as the velocity at the moment when the end tips of escaping legs reach zero depth position ($t = t_t$, and $H(t_t) = 0$).

**Modes of jumping.** We observed several cases in which a leg quickly sank under the water surface after the distal end of the leg pierced the meniscus during the stroke at the average depth of 3.7 mm (Fig. 3c), which is close to the sinking depth of a long thin rigid cylinder $\sqrt{2}l_c$ (3.8 mm for water) in a quasi-static condition[14,15] (Supplementary Fig. 4 and Supplementary Note 4). For a relatively long maximum downward reach of legs ($L > \sqrt{2}$), the excessive angular leg velocity leads to the dimple depth deeper than the sinking depth $\sqrt{2}l_c$ and penetration of water surface; this mode of jump is referred to as the meniscus breaking jump. The jumps that do not involve meniscus breaking can be theoretically categorized into two types: pre-takeoff closing and post-takeoff closing jumps, depending on when the legs are fully closed. In the pre-takeoff closing jumps, the legs complete their rotation before leaving the water surface ($\omega t_c = \pi/2$ and $H(t_c) > 0$), whereas in the post-takeoff closing jumps, the legs are fully rotated in the air after

takeoff ($\omega t_t < \pi/2$ and $H(t_t) = 0$). Here, $t_c$ indicates the instant of the end of leg closing movements. Only two of these modes of jumping, the post-takeoff closing and meniscus breaking jumps (surface breaking by one or two legs) were observed in water striders as described in Fig. 3. This classification is important for model predictions (Fig. 4; see below).

**The optimal jump and test of the model predictions.** When we solved equation (3) and used the parameters extracted directly from the videos of jumps of the actual individual water striders, the model reasonably predicted the observed maximum dimple depth (Supplementary Note 5 and Supplementary Fig. 5a). When we used the parameters extracted from the videos to calculate the takeoff velocity of a water strider (via integrating the instantaneous net force on the body over time until the tip of leg reached the zero depth position, $t = t_t$), the results reasonably agreed with

the empirically measured takeoff velocity (Supplementary Note 6 and Supplementary Fig. 5b,c). These calculations indicated that the model correctly approximates the physical processes involved in jumping.

Using the model, we derived the theoretical predictions about the optimal vertical jumping behaviour assuming that predator-mediated natural selection maximizes the vertical takeoff speed. Fast vertical takeoff is important for survival because it quickly removes the insect from the vicinity of the approaching predators such as fish and backswimmers, which attack upwards from under the water surface[6,7]. We compared the theoretical predictions with empirical data from slow motion videos of five species of water striders with different body size and leg morphology: *G. remigis*, *G. comatus*, *G. latiabdominis*, *G. gracilicornis* and *A. paludum* (body mass and leg morphology are described in Supplementary Table 1; see Methods section for details).

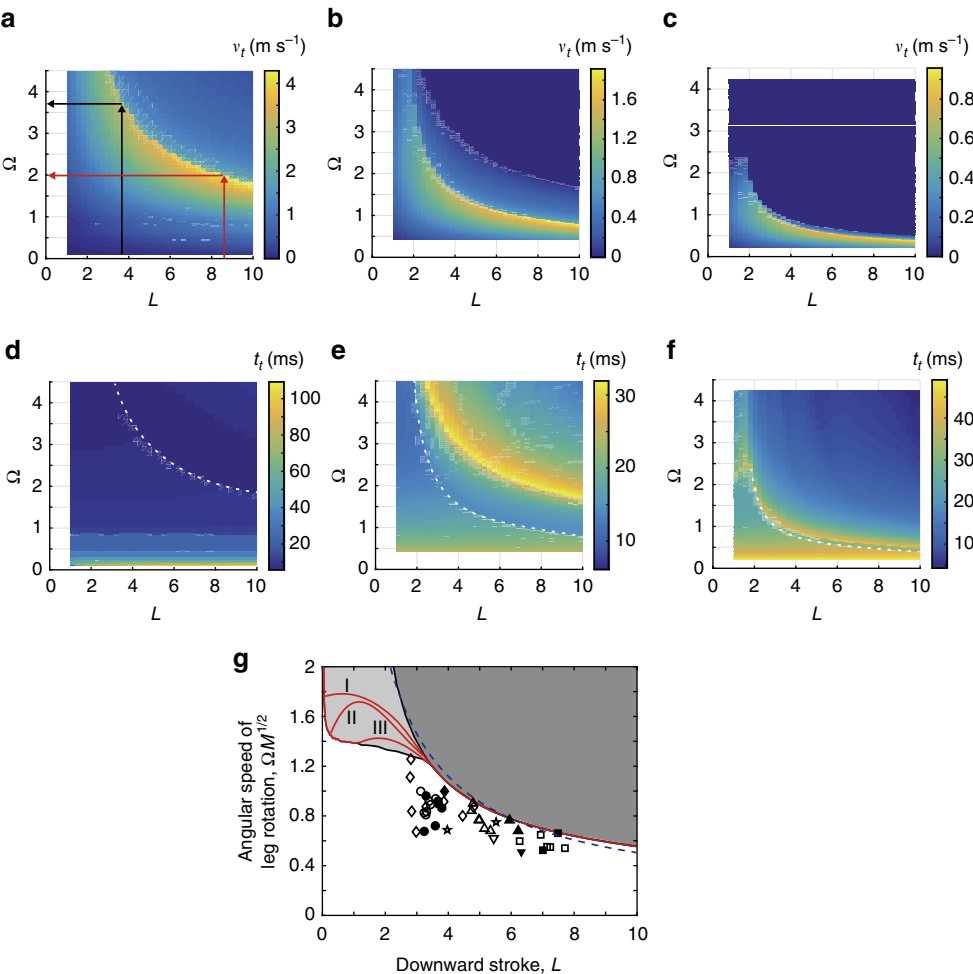

**Figure 4 | Theoretical and empirical results of jumping of water striders.** (**a**–**c**) Effect of the dimensionless angular velocity of the leg $\Omega$ and the dimensionless maximum downward reach of leg $L$ on takeoff velocity $v_t$ with warmer colours indicating higher takeoff velocity for water strider species of three different sizes expressed in different values of the variable $M$: $M = 0.1$ in **a**; $M = 0.5$ in **b**; $M = 2.0$ in **c**. $M$ represents the ratio of body mass to the maximal mass of water that the legs can displace by pushing against the water surface. Water striders observed in this study have $M$ near 0.5. The arrows in **a** show that the longer leg indicated by red arrows should move slower, for example, than the leg indicated black arrows to get maximum takeoff speed. (**d**–**f**) Effect of the dimensionless angular velocity of the leg $\Omega$ and the dimensionless maximum downward reach of legs $L$ on the time taken to escape from water $t_t$ with warmer colours indicating longer escape time corresponding to the conditions of **a**–**c**, respectively. White dashed lines indicate the boundary of meniscus breaking jump. (**g**) Phase diagram for the three jump modes as a function of $\Omega M^{1/2}$ and $L$: post-takeoff closing (white area), pre-takeoff closing (light shaded area), and meniscus breaking (dark shaded area). The red lines marked with I, II and III indicate the conditions resulting in maximal vertical takeoff velocity with three $M$ in **a**–**c**: I with $M = 0.1$; II with $M = 0.5$; III with $M = 2.0$. The dashed line shows the line of $\Omega M^{1/2} = 4/L + 0.1$. The phase diagram includes empirical results from the jump characteristics of females (filled symbols) and males (unfilled symbols) of *G. remigis* (inverted triangles), *G. comatus* (diamonds), *G. latiabdominis* (circles), *G. gracilicornis* (triangles) and *A. paludum* (squares) with nymph of *G. remigis* (stars).

First, we calculated the predicted takeoff speed as a function of three variables that can be derived from empirical measurements on insects: the dimensionless angular velocity of the leg $\Omega$ (which is directly related to the angular velocity of leg rotation $\omega$, which in turn can be calculated from empirically measured maximum downward leg velocity, $v_{s,\max}$, and the maximal reach of legs, $\Delta l$, according to the formula $\omega = v_{s,\max}/\Delta l$), the dimensionless maximum reach of leg $L$ (which is directly correlated with leg length), and the variable $M$ (Fig. 4), representing the body mass expressed in units of the maximal mass of water that the tibia plus tarsus can displace. Hence, the model allowed us to predict the combinations of leg morphology $L$ and behaviour $\Omega$ that result in the maximum takeoff speed (narrow yellow areas in Fig. 4a–c) and corresponding time to escape from water (Fig. 4d–f) for insects of different sizes $M$.

For a given $M$, the range of $L$ and $\Omega$ values that maximize takeoff speed is relatively narrow (Fig. 4a–c). The optimal value of $\Omega$ tends to decrease with the increase of $L$. That is, if we consider an insect of a specific $M$ (that is, an insect of specific mass $m$ and tibia plus tarsus length $l_t$), we expect that the longer is the femur (resulting in longer $L$), the slower should be the leg movements to produce the optimal jump (slower $\Omega$). For insects with large mass and/or short tibia plus tarsus length $l_t$, resulting in relatively large $M$ (example in Fig. 4c; $M = 2.0$), the values of $\Omega$ that may produce the maximal takeoff speed are relatively low (in Fig. 4c). However, for a typical water strider with relatively long tibia plus tarsus $l_t$ and small mass resulting in small $M$ (for example, $M = 0.5$ in Fig. 4b), the leg rotational speeds that may produce optimal jump are relatively large (in Fig. 4b), and a water strider should be able to precisely adjust its leg rotation $\Omega$ to its leg length $L$ in order to produce the optimal jump. For example, if an insect of $M = 0.1$ (Fig. 4a) has long legs $L$, then its optimal leg rotation speed $\Omega$ should be low (red arrows in Fig. 4a). But, if an insect of the same $M$ had short legs, then its optimal leg rotation would be fast (black arrows in Fig. 4a). The insects should be careful to not go over the optimal leg rotation speed because it may result in piercing of the water surface and a sudden decrease in jump performance (notice sharp transition from yellow to blue especially for large values of $L$; Fig. 4a–c) as the leg rotation $\Omega$ increases (also see Supplementary Fig. 5b and Supplementary Note 6). Moreover, Fig. 4d–f shows that the optimal conditions maximizing takeoff velocity also minimize the time to escape from the water surface for post-takeoff closing jump, which may increase the insects' survival rate (an alternative 3D graphical representation of Fig. 4a–f shown in Supplementary Fig. 6 and Supplementary Note 7).

Finally, to compare the theoretical predictions with corresponding empirical data from water striders of different sizes $M$, we constructed a two-dimensional regime map for the three modes of jumps in the space of two dimensionless variables derived from equation (3): the leg length $L$ and the composite variable $\Omega M^{1/2}$ involving leg rotation, body mass and tibia plus tarsus length (Fig. 4g). Because variation in $\Omega M^{1/2}$ expresses mostly the variation in $\Omega$ rather than $M$ in real striders, the variable $\Omega M^{1/2}$ mostly represents the behavioural trait $\Omega$, justifying our approximate view of $\Omega M^{1/2}$ as a behavioural index (see Supplementary Note 8 and Supplementary Fig. 7). The critical line of meniscus breaking could be simply approximated as $L \sim \Omega^{-1} M^{-1/2}$ (the blue dashed line in Fig. 4g) by balancing velocity of body centre $v$ of an insect and average downward velocity of the four legs with respect to the horizontal plane through body centre $v_s$ at $t = t_m$ (for the derivation, see Supplementary Note 9). The red lines in Fig. 4g, corresponding to the condition for the maximal takeoff velocity, are located in the area of pre-takeoff closing jump for insects with relatively short legs, that is, when the maximum downward reach of legs

$L \leq 3.5$. But, for long-legged insects, with $L > 3.5$ typical for many water strider species, the fastest jump occurs when the insect drives its legs at the speed just below the meniscus breaking condition. We found that the jumping of water striders occurs always near the condition for the maximal takeoff velocity as shown in Fig. 4g. Most jumps occurred just a little below the critical line of meniscus breaking (Fig. 4g) as if the animals kept a certain safety margin to avoid the breaking of water surface that may dramatically decrease their chances of successful escape from predators.

## Discussion

The reasonable match between the model predictions and the empirical findings for the maximal takeoff speeds suggests that water striders have the ability to adjust their behaviour (angular leg movements) to reach the optimal conditions for the fastest jump away from danger. The morphological traits appearing in our model, that is, body mass $m$, tibia plus tarsus length $l_t$, whole leg length $l_l$ and leg radius $r$, are not likely to have been optimally designed for only a single particular function, such as jump escape. However, the behavioural trait can in principle be adjusted by individuals within the animal's physical abilities. Based on our results, we hypothesize that a water strider of a given mass and morphology may control its stroke speed by modifying the leg rotation's angular velocity to attain the maximum jumping speed as an adaptation to avoid predation. This then leads to a question of whether the hypothetical optimal adjustment of the leg movements to morphology is achieved by natural selection for a 'hardwired' species-specific motor pattern or by individual learning. Our model enables the pursuit of a variety of similar questions in the future because, in principle, it may be used to predict the effect of leg angular movements as well as morphological features on jump performance.

For simplicity, we only focused on the vertical jumping with synchronized leg locomotion. In those jumps, the dimples on the water surface under mid-legs occur approximately simultaneously with the dimples under hind-legs[11], justifying our simplifications for modelling purposes. Because in our model the insect body mass is located in one point of space, the 'body centre', we also ignored the distinction between downward movements of tibia and tarsus due to the actual leg being pushed away from the body centre by muscles and to the rotation of the insect's body axis during jump. The latter appears to contribute to the hind leg pushing against the water surface during the pushing stage of jump when body axis changes from horizontal to about 45° by the moment of maximal dimple depth (Fig. 1b).

Despite its simplicity, our model is sufficient to provide us crucial insights into water strider's near-vertical jumping on water. The fast upward jump is the best solution to escape predators[6,7] in those situations when insects are surprised by an attack from under the water surface, while the attacker's approaching movement trajectory cannot be tracked by prey. Actually, the water striders also may move their four legs in many different ways, showing a variety of jumping trajectories and speeds, including back somersaults and an apparent ability to control their jump trajectories with respect to the direction of the approaching danger in situations when they can perceive it. The general theoretical approach in this study would be still valid for water striders' upward jumping with four legs moving differently if we build complex locomotion functions to reflect the four leg's motions of each case.

Together with Koh et al.[11], our results prove that water striders are able to exploit water surface properties to optimally perform their predation avoidance jumps using elongated legs that are developmentally shaped by recently discovered genetic mecha-

nism[16], which is an apparent outcome of natural selection to avoid predator attacks. This work adds to the documented or suggested repertoire of water striders' behaviours that evolved to exploit the water surface properties: defence of territories[17–21], courtship[21–23], establishing dominance in agonistic interactions[20,24], defence of mates against harassment from other males during mate guarding[25], sex recognition[20,21,26] and sensing distribution of food in environment[27] or possibly also sensing local sex ratio in a population[28].

In summary, our study provides a mathematical understanding of how a biological organism may achieve the optimal level of motility on water surface by apparently tuning the behavioural trait to its morphology. Whether species-specific leg movements are innate and shaped by co-evolution with morphology, or behaviourally plastic and shaped by individual experience during jumping, remains to be determined. Many studies have not directly determined the role of individual learning in adjusting the locomotion behaviour to locomotory organs' morphology, but they generally documented a similar match between the morphology and behaviours in a variety of mostly vertebrate taxa[29–39]. The actual tuning by individuals of their locomotory behaviour to morphology has been documented in only some organisms. For example, juvenile *Acrocephalus* birds are able to learn from experience and adjust the use of perching sites and habitats in order to optimally tune the use of legs for perching to their legs' morphology[40,41]. It is possible that the water striders may also be able to adjust their leg movement to changes in physical conditions of jumping, highlighting the possibility that insects also are able to adjust their behaviour to morphology through individual locomotory learning.

The model opens new avenues of research. Biologists can start with our model to ask questions on the evolutionary mechanisms that shape jump-optimizing morphology and behaviour of water striders with known phylogenies[42–44]. Modified model can also be used to understand jumping of other insects of similar ecology, such as springtails or fishing spiders that exploit capillary force to jump from water. The fundamental concepts presented in this study can also give a guideline to develop semi-aquatic robots that aim to emulate the superior locomotory abilities of the water striders on water[45–47].

## Methods

**Experimental setup.** For slow motion filming we used three Asian species (*G. latiabdominis*, *G. gracilicornis* and *A. paludum* from streams and ponds around the Seoul National University, and from ponds in Yongsan area, Seoul, Korea), and two North American species (*G. remigis*, *G. comatus* from the Huyck Preserve, NY, USA). Insects were filmed in the lab using artificial lighting (Korean species) or in the outdoors using sunlight (North American species). A water strider was induced to jump in a square acrylic bath (70 mm wide) half-filled with water, and two high-speed cameras (Trouble Shooter 1000 ME; Fastec Imaging Inc., San Diego, CA, USA) were used to record the jumping behaviours at 500 or 1,000 frames per second from the front and side views simultaneously. A total of 39 jumps by 30 adult water striders and three jumps by two nymph water striders (morphology described in Supplementary Table 1) were recorded and analysed, where the inclinations of jump trajectory were between 60° and 80° to the horizontal with almost bilateral symmetry with respect to the leg of distortion of the water surface. Most of the jumps were stimulated by a rigid stick positioned below the water strider's body, and we only analysed the jumps verified that the insect's body or the leg was not pushed upwards by the stick. Fresh body mass of an insect was measured right after filming. Each animal was also photographed (a ruler was present in each photo to provide the scale) and the lengths of leg segments were measured from the photographs using ImageJ (http://imagej.nih.gov/ij/)[48]. The animals used in the present study were handled in accordance with institutional guidelines for the care and use of laboratory animals. Korean law does not require special permits for the use of water striders in research. The Huyck Preserve permitted us to use the water striders on their private land.

**Model.** Matlab was used to obtain the results described in Fig. 4, Supplementary Figs 5,6. To solve equation (3), ode15s function was used.

**Code availability.** The code is available from authors on request.

**Data availability.** All relevant data are available from authors on request.

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

## Acknowledgements

This work was supported by the National Research Foundation of Korea Grants 2014023206 and NRF-2013R1A2A2A01006394, Disaster and Safety Management Institute Grant MPSS-CG-2016-02, and Bio-Mimetic Robot Research Center funded by Defense Acquisition Program Administration under Grant UD130070ID. The following persons helped in analysing the videos of jumping water striders: S. Song, C.K. Kang, J. Moon.

## Author contributions

E.Y., S.-i.L., P.G.J. and H.-Y.K. designed the study. J.H.S., S.-i.L. and P.G.J. conducted experiments on water striders, extracted data and provided the results to E.Y., who analysed the jumping dynamics from these results and built the theoretical model. E.Y., S.-i.L., P.G.J. and H.-Y.K. discussed the results and the model, and provided feedback for model development and modifications by E.Y. E.Y. wrote the initial manuscript, which was subsequently modified, after detailed input from, and discussions with, P.G.J., S.-i.L. and H.-Y.K.

## Additional information

**Competing financial interests:** The authors declare no competing financial interests.

**Publisher's note**: 

