## [Peer Review File · Nature Communications]

Reviewers' comments:

Reviewer #1 (Remarks to the Author):

The paper written by Yang et al. is a solid and interesting contribution to animal locomotion at interfaces, for which we know comparatively much less. The experimental portion is of good quality and the modeling, in particular the dimensional analysis, very nice. While this study is basically a follow-up of their publication (ref # 9) I support its publication once the major points have been addressed. The following comments address mainly the context in which the study was conducted and hence the implications, but do not put the core results into jeopardy.

MAJOR COMMENTS

A. Biology. The antipredator response of jumping in water striders is assumed, but the support for it is very weak, and translates in the fact that not a single reference for this assumption could be given. In fact, one may wonder whether jumping of a few centimeters is an efficient strategy against fishes which can jump out of water over much larger distances. By contrast, a lot is known about water striders using wave trains to communicate sexually (see the papers of Wilcox, some of their own and others). Using water surface as a trampoline is therefore known to be of use and the authors should look into this aspect of the biology to get a more solid foundation for the evolutionary basis of their work. The same applies somewhat to their statements about morphology and behavior being nicely linked. I find it not surprising that organisms use their own appendages to their best use; the two aspects "co-evolved" concurrently. If the authors believe that the link is especially strong, they would need to compare it to other similar links between locomotion traits in other organisms and give references for the claim.

B. Modelling. All four legs are assumed to work in the same fashion, by using a mean value. I am not quite convinced. The legs are not of all the same length, so they cannot reach the same point in space during their course, or apply the same force. Their kinematics is also somewhat different too (see the paper cited in the reference, 6). I am unclear whether taking such statistical approach is OK or not and why.

C. Physics . Some key aspects of figure 4 come about in a way which could have been explained more intuitively, with reference to the Weber number. The maximum harvestable energy from the water surface is by staying just below $We=1$, or about $1=\rho U^2 \cdot \text{length}/\text{surfacetension}$. This

can be rewritten as $U \sim \sqrt{1/\text{length}}$, leading to the plotted relationship. Reference 9 was already giving this approach.

D. Physics. As you say, most jumps occurred just a little below the critical line of meniscus breaking. However, you seem to condition the breaking of the surface at the maximal depth, and not as function to the ratio of forces. Why not and would it differ?

E. Biology. Some animals are able to jump despite breaking the surface. Is this because only a few legs do so, or is there a more complex hydrodynamical transient interaction which still produce some lift, or is it due to the timing of events ? More generally, finding the ratio of forces within the main text rather than in the appendices would be good. Finally, related to this is the following question: how do we reconciliation the fact that the ratio of inertial forces vs capillary forces is of the order of 10^{-4} , and that there are still meniscus breaking jumps?

E. Measurements and physics. It is worth restating that the measurements are done with a single vertical plane of "light". The dim under each leg is not isotropic, because the length of tibia+tarsus is much more than the diameter. So, for horizontal movement (as mentioned in the appendix and end of the discussion) the interplay between forces acting on horizontal and vertical planes will be more complex. In other words, the work cannot be translated as such to horizontal locomotion.

MINOR POINTS

1. Abstract. Delete "dramatic" and "the" on line 25.
2. Abstract and conclusion. One may wonder (and some people do even write this very publicly) what kind of tasks such microrobots will be able to carry. Certainly NOT pollution surveillance, there are plenty of less costly ways to do so. While such hype seems to fly when writing grants, I would tone it down in publications.
3. Results. Can you give us an information about how many insects did jump with angle smaller than 60° ?
4. The model was run with $M=0.1, 0.5$ and 2 , but I wish you would do $M=1$, as it seems to be of special interest
5. lines 280-282 are a repeat of previous text.
6. Legend figure 1. Are you implying that the velocity profile in (f) , being from Koh et al., is not from the same dataset ? if so, how can you use that ?
7. Figure 4. I am not sure Figure (g) is a test of the model. It seems to be rather an overlay of the observed data points on the space defined by the two adimensionalized variables. Supplementary Figure 4 is by contrast a test.

8. Legend supplementary Figure 4. I do not understand the implication and rationale of the last sentence, about the overestimation of velocity. Please expand.

9. Supplementary figure 6 seems to be unused?

10. A graphic showing all the distances, found in supplementary table 1, would be very handy.

11. How do you measure Δt ?

Jérôme Casas

Reviewer #2 (Remarks to the Author):

Yang et al. present a study on jumping behaviors of water striders from the water surface. They found that their jumping kinematics are chosen to be optimal based on maximizing the vertical take-off speed. In particular, the authors looked at the leg rotation and length to reach the maximal speed using the capillary force. In general, this manuscript is well written and has a lot of scientifically interesting phenomena and analyses. I do recommend this manuscript for publication after revisions based on the comments below.

Major comments:

o There are a number of forces neglected in the force balance. 1) viscous or inertial drag while the leg moves along the water surface. 2) surface tension on the contact line while the leg is pulled out from the dimple. 3) Hydrostatic pressure from an air pocket formed by the leg. These forces might be small, which this reviewer is also expecting. However, it would be good to show some non-dimensional numbers or the order of magnitude calculations of these forces before the authors introduce the force (or momentum) equation.

o All analyses were done in 2D projected plane. This reviewer is sure the authors already considered 3D effect, but it is not clear in the current manuscript. All measured lengths and rotation angle are in the 2D projected plane or 3D?

o Measured angular velocity has some variations in insect data as shown in Figure 2c, which is reasonable since it is computed from animal data. So, insect data in Figure 4g should have similar fluctuations in the measurement in angular speed of leg rotation (y-axis). The authors need to show how much variations data points in figure 4g have.

Minor points:

o In Figure 1, l_v is shown instead of l_s

o There are so many length scales for the leg (l_t , l_l , l_w). This reviewer wonders whether the authors provide a simple schematic of the water strider to show how different lengths are defined.

[Reviewer 1]

MAJOR COMMENTS

Comment A-1:

Biology. The antipredator response of jumping in water striders is assumed, but the support for it is very weak, and translates in the fact that not a single reference for this assumption could be given. In fact, one may wonder whether jumping of a few centimeters is an efficient strategy against fishes which can jump out of water over much larger distances.

Response:

Thank you for the comments and suggestions. In accordance with the suggestion of the reviewer we have added citations concerning escape behaviour as antipredatory strategy. The following papers are now mentioned in the rewritten parts of the discussion (line 29-31, 309).

Krupa, J. J. & Sih, A. Comparison of antipredator responses of two related water striders to a 373 common predator. Ethology 105, 1019-1033 (1999).

Haskins, K., Sih, A. & Krupa, J. Predation risk and social interference as factors influencing habitat selection in two species of stream-dwelling waterstriders. Behav. Ecol. 8, 351 (1997).

Armisen, D., Refki, P. N., Crumiere, A. J. J., Viala, S., Toubiana, W. & Khila, A. Predator strike shapes antipredatory phenotype through new genetic interactions in water striders. Nat. Commun. 6, 8153 (2015).

Although none of these papers, or any other paper we are aware of, evaluate efficiency of escapes from fishes in natural situations, the fact that water striders perform these escape jumps in response to attacks from under water suggests that they do provide sufficient protection, and mediate natural selection maintaining this behaviour in water striders. In this context one has to remember that the jumps in natural situation are often performed in a quick series and this undoubtedly contributes to their efficiency as escape behaviour. Efficiency to escape from predatory attacks have been measured in artificial conditions (e.g. the newest paper by Armisen et al.¹⁶) and has been shown to contribute to avoidance of capture by predators.

Comment A-2:

By contrast, a lot is known about water striders using wave trains to communicate sexually (see the papers of Wilcox, some of their own and others). Using water surface as a trampoline is therefore known to be of use and the authors should look into this aspect of the biology to get a more solid foundation for the evolutionary basis of their work.

Response:

In accordance to the reviewer's suggestion we have added text to the discussion (line 308-313), where we cite some of the research on how water striders exploit water surface tension properties for communication. These are the new references that we are adding to the manuscript now:

- Wilcox RS, Spence JR. 1986. The mating system of two hybridizing species of water striders (Gerridae). I. Ripple signal functions. *Beh. Ecol. Sociobiol.* 19: 79-85.
- Vepsalainen K, Nummelin M. 1985b. Male territoriality in the water strider *Limnioporus rufoscutellatus*. *Annales Zoologici Fennici* 22:441-448.
- Vepsalainen K, Nummelin M. 1985a. Female territoriality in the water striders *Gerris najas* and *G. cinereus*. *Annales Zoologici Fennici* 22:433-439.
- Jablonski PG, Vepsalaäinen K. 1995. Conflict between sexes in the water strider, *Gerris lacustris*: a test of two hypotheses for male guarding behavior. *Behav. Ecol.* 6: 388-392.
- Jablonski PG, Wilcox SR. 1996. Signalling asymmetry in the communication of the water strider *Aquarius remigis* in the context of dominance and spacing in the non-mating season. *Ethology* 102:353-359.
- Wilcox RS, Stefano JD. 1991. Vibratory signals enhance mate-guarding in a water strider (Hemiptera: Gerridae). *Journal of Insect Behavior* 4: 43-50.
- Jablonski PG, Scinski M. 1999. Water striders are prescient foragers: use of sensory information for patch assessment in food-based territoriality of *Aquarius remigi* (Gerridae, Heteroptera). *Polish Journal of Ecology* 47:247-256.
- Han CS, Jablonski PG. 2010b. Role of body size in dominance interactions between male water striders, *Aquarius paludum*. *J Ethol* 28:389-392.
- Han CS, Jablonski PG. 2010a. Male water striders attract predators to intimidate females into copulation. *Nature Communications* 1, 52: doi:10.1038/ncomms1051.
- Wilcox RS. 1979. Sex discrimination in *Gerris remigis*: role of surface wave signal. *Science* 206: 325-327.
- Han CS, Jablonski PG. 2009. Female genitalia concealment promotes intimate courtship in a water strider. *PLoS ONE* 4(6): e5793. doi:10.1371/journal.pone.0005793
- Jablonski PG. 1996. Intruder pressure affects territory size and foraging success in asymmetric contests in the water strider *Gerris lacustris*. *Ethology* 102: 22-31.

Comment A-3:

The same applies somewhat to their statements about morphology and behavior being nicely linked. I find it not surprising that organisms use their own appendages to their best use; the two aspects "co-evolved" concurrently. If the authors believe that the link is especially strong, they would need to compare it to other similar links between locomotion traits in other organisms and give references for the claim.

Response:

In accordance with the comment from the reviewer we added citation of papers that also study associations/correlations between morphology and behaviour, and we have added several sentences on this subject to the discussion (line 314-328) on the revised manuscript.

Norberg UM. 1979. Morphology of the wings, legs, and tail of three coniferous forest tits, the goldcrest and the tree creeper in relation to locomotor pattern and feeding station selection. *Philosophical Transactions of the Royal Society of London* 287: 13 1-1 65

Webb PW. 1984. Body form, locomotion and foraging in aquatic vertebrates. *Am Zool* 24:107-120.

Norberg UM, Rayner JMV. 1987. Ecological morphology and flight in bats (Mammalia: Chiroptera): wing adaptations, flight performance, foraging strategy and echolocation. *Philosophical Transactions of the Royal Society of London, Series B, Biological Sciences*, 316:335-427.

Losos JB. 1990. The evolution of form and function: morphology and locomotor performance in west indian Anolis lizards. *Evolution* 44:1189-1203.

Moreno E, Carrascal LM. 1993. Leg morphology and feeding postures of four Parus species: an experimental ecomorphological approach. *Ecology* 74:2037-2044.

Gerstner CL. 1999. Maneuverability of four species of coral-reef fish that differ in body and pectoral-fin morphology. *Can J Zool* 77:1102-1110.

Brana F. 2003. Morphological correlates of burst speed and field movement patterns: the behavioural adjustment of locomotion in wall lizards (*Podarcis muralis*). *Biol J Linn Soc* 80:135-146.

Dial KP. 2003. Evolution of avian locomotion: correlates of flight style, locomotor modules, nesting biology, body size, development and the origin of flapping flight. *Auk* 120:941-952

Stiles FG, Altshuler DL, Dudley R. 2005. Wing morphology and flight behavior of some north American hummingbird species. *Auk* 122:872-886.

Brewer ML, Hertel F. 2007. Wing morphology and flight behavior of Pelecaniform seabirds. *Journal of Morphology* 268:866-877.

Tytell ED, Borazjani I, Sotiropoulos F, Baker V, Anderson EJ, Lauder GV. 2010. Disentangling the functional roles of morphology and motion in the swimming of fish. *Integr. Comp. Biol.* 50:1140-1154.

Ley, H.-W. 1988. Verhaltensontogenese der Habitatwahl beim Teichrohrsinger (*Acrocephalus scirpaceus*). *J. Orn.* 129: 287-297.

Leisler B, Ley HW, Winkler H. 1989. Habitat, behavior and morphology of *Acrocephalus* warblers: an integrated analysis. *Ornis Scandinavica* 20:181-186.

Comment B:

Modelling. All four legs are assumed to work in the same fashion, by using a mean value. I am not quite convinced. The legs are not of all the same length, so they cannot reach the same point in space during their course, or apply the same force. Their kinematics is also somewhat different too (see the paper cited in the reference, 6). I am unclear whether taking such statistical approach is OK or not and why.

Response:

In accordance with the reviewer's suggestions we commented on the assumptions made in our model in the revised Supplementary Note 1.

The whole model is an abstraction, a simplification, and using average angular velocity of legs (angular with respect to a horizontal plane and not body axis) is a part of this simplification which we abstract from the real kinematics of leg movements in relation to body, and from considering the body pitch changes by assuming body mass in the one point in space. We think that this is reasonable for vertical and near-vertical jumps. Those jumps have been apparently not studied. Even the jumps studied by Caponigro and Eriksen⁸, which appear to represent “leaps forward” (Fig. 5 and Fig. 7 of their paper), showed reasonable similarity in the leg movements (shown by little arrows in figures in their paper) between hind and mid-legs, unlike for “rowing”, for which clear differences in leg movements have indeed been shown by these authors. Nevertheless, most of their analyses were only on the bases of the view from above and therefore they represent the horizontal angular movements rather than the vertical ones.

Our own measurements of dimple depths dynamics show that upward force from dimples created by the hindlegs may be slightly smaller than the upward force from the dimples of the midlegs. But, as Koh et al.¹¹ showed in Fig. 1A (left column of frames), and we show in the current manuscript in Fig. 1f, the dimple depth changes simultaneously for both middle and hind legs, indicating that the middle and hind legs reach and interact with water surface in a manner not so different from each other. In this situation the use of an average upward force from all four legs is not much different from using the separate upward force from hindlegs’ dimples and midlegs’ dimples. In order to precisely address the reviewer’s comment, we have inserted Supplementary Fig. 1b and the second paragraph explaining it in the Supplementary Note 1, which shows the calculated ratio of force obtained by using mean values of wetted length and dimple depth of middle and hind legs to force obtained by using different values of middle and hind leg, as a function of wetted length ratio and dimple depth ratio of middle leg to hind leg. The results show the validity of our simplification and imply that the simplification with mean values of wetted length and dimple depth of middle and hind legs is reasonable.

Here, we attached the inserted paragraph and Figure in the Supplementary Note 1.

In addition, we used average values of wetted length and resulting dimple depth made by middle and hind legs. We exploit this simplification because equations of motion become tractable and the resulting theoretical predictions are accurate enough. Supplementary Fig. 1b shows the verification of this simplification. The color map indicates the ratio of two forces; fourfold of the force \bar{F} calculated with mean values of the wetted length \bar{l}_w and dimple depth \bar{h} of middle and hind legs and the sum of the forces on the four legs with different values of the wetted length and dimple depth of middle and hind legs,

$$\frac{4\bar{F}}{\Sigma F} = \frac{4\bar{l}_w\bar{h}\left[1-(\bar{h}/2l_c)^2\right]^{1/2}}{\Sigma l_w h\left[1-(h/2l_c)^2\right]^{1/2}}.$$

The black dots show the measured value from jumping of water striders we observed when the legs reach the deepest position. The observed conditions have force ratios between 0.76 and 1.15 implying that our simplification is reasonable, except for the three cases with the highest dimple depth ratio, where the maximum dimple depths made by hind legs were below 1 mm and the resulting force ratio about 0.65.

Supplementary Figure 1. Validation of synchronous motion of four legs (a) Comparison of the moment of maximum depth of dimple generation t_m between middle and hind legs. The correlation in each trial results in the correlation coefficient $r = 0.943$, p -value = 0.0311, and $df = 28$ implying the synchronous motion of four legs. Data from the jump of females (filled symbols) and males (unfilled symbols) of *G. latiabdominis* (circles), *G. gracilicornis* (triangles), and *A. paludum* (squares) are plotted. Dashed line indicates the exact match between middle and hind legs, and solid line the fitted regression line. (b) The ratio of the forces calculated with mean values of the wetted length and dimple depth of middle and hind legs to the force with different values of the wetted length and dimple depth of middle and hind legs, as a function of the ratio of wetted lengths and dimple depths made by middle and hind legs. The black dots indicate the observed jumps of water striders.

Comment C:

Physics. Some key aspects of figure 4 come about in a way which could have been explained more intuitively, with reference to the Weber number. The maximum harvestable energy from the water surface is by staying just below $We = 1$, or about $1 = \rho U^2 l / \sigma$. This can be rewritten as $U \sim \sqrt{1/l}$, leading to the plotted relationship. Reference 9 (or reference 11 in the revised manuscript) was already giving this approach.

Response:

Considering the reviewer's comment, we have suggested a simple relationship guiding the meniscus breaking boundary of jumping on water. To achieve maximum takeoff velocity, it is important to fully push the surface downward quasi-statically to maximize momentum obtained from capillary force. The Weber number could not give the exact boundary of meniscus breaking jump, but We being much smaller than 1 indicates that capillary force is dominant. As the reviewer suggested, we have found an intuitive relationship $L \sim \Omega^{-1} M^{-1/2}$ simplifying the meniscus breaking boundary as shown in Fig. 4g. We have derived this relationship from the balance between vertical velocity of body centre v and average downward velocity of the four legs with respect to the horizontal plane through body centre v_s . This relationship is inserted in the middle of section [The optimal jump and test of the model predictions], in lines 255-258 in the main text, and detailed derivation of the relation is given in the Supplementary Note 9.

Comment D:

Physics. As you say, most jumps occurred just a little below the critical line of meniscus breaking. However, you seem to condition the breaking of the surface at the maximal depth, and not as function to the ratio of forces. Why not and would it differ?

Response:

Koh et al.¹¹. emphasized that, to jump efficiently by using capillary force like water striders, the Weber number of the driving motion of leg should be much smaller than 1. This condition implies that the dynamic effects can be neglected and the surface acts as a membrane unless the leg pushes down the surface below the limit depth the capillary force can bear. Here, capillary force on a leg is a function of the depth, so the maximum depth limit is equivalent to the maximum force limit. Because Koh et al.¹¹. aimed at building a robot, the driving force of robot and the maximum force limit were important, and they considered the maximum force limit. On the other hand, this study tries to model water striders' motion and analyze their jumping performance, therefore, we use the depth limit.

We add a comment on this equivalency of these two approaches at the end of the Supplementary Note 4, as *“In addition, we note that the maximum depth limit is equivalent to the maximum force limit¹¹, or the force per unit wetted length f should satisfy $f < 2\sigma$, because capillary force on a leg is determined by the dimple depth^{13, 15}.”*

Comment E-1:

Biology. Some animals are able to jump despite breaking the surface. Is this because only a few legs do so, or is there a more complex hydrodynamical transient interaction which still produce some lift, or is it due to the timing of events?

Response:

In accordance with the reviewer's comment we explain here the variability of mechanisms of jump on water surface. Several arthropods use different jumping mechanisms on water. Springtails have been known to exploit capillary force for jumping on water as described by Hu et al. (Experiments in Fluids, 43(5), 769-778, 2007). To jump on water, a springtail releases its spring-loaded tail and pushes down the water surface. The body surface is superhydrophobic and the 1 mm long tail generates stroke with Weber number much less than 1. This short tail cannot penetrate (or break) the water surface and it demonstrates very efficient jumping on water.

Whereas, it has been reported that pygmy mole crickets jump off water by using viscous friction by Burrows and Sutton (Current Biology, 22(23), R990-991, 2012). During jump, they extend their hind legs rapidly and the legs penetrate the water. This extension induces flaring of the paddles and spurs on the legs that act as oars under water. In this case, the laminar flow around the submerged legs, paddles, and spurs generates the viscous friction-based thrust instead of the capillary force.

Fishing spiders mainly use pressure drag as reported by Suter and Gruenwald (Journal of Arachnology, 28(2), 201-210, 2000). Although the locomotion seems to be similar to that of water striders, the length and speed of legs is much larger than water striders, implying different mechanisms between two species. Spiders' legs usually reach much deeper than the critical depth of meniscus breaking and still keep the cavities around the legs, and thus they generate pressure drag and water flows downward.

Among these different mechanisms, meniscus breaking should be considered only when the dominant thrust originates from capillary force. Therefore, only the jump exploiting capillary force with longer leg than the critical depth of meniscus breaking would be affected by meniscus breaking.

Comment E-2:

More generally, finding the ratio of forces within the main text rather than in the appendices would be good. Finally, related to this is the following question: how do we reconcile the fact that the ratio of inertial forces vs capillary forces is of the order of 10^{-4} , and that there are still meniscus breaking jumps?

Response:

In accordance with the reviewer's suggestions, we have moved the paragraph explaining the ratio of forces from Supplementary materials to the section [Theoretical model], in lines 82-97, in the main text. Meniscus breaking occurs when a dimple depth made by a thin cylinder or a leg exceeds the critical depth that the capillary force can bear. Legs can reach the critical depth even when $We \ll 1$ as it slowly pushes down the water surface. This is totally different from splashing where $We > 1$, or the relatively large amount of kinetic energy of the cylinder or the leg hitting the water surface spatters water. We have inserted the explanation clarifying that meniscus break can be caused by capillary forces in a quasi-static situation, in line 185 in the section [Modes of jumping].

Comment F:

Measurements and physics. It is worth restating that the measurements are done with a single vertical plane of "light". The dimple under each leg is not isotropic, because the length of tibia+tarsus is much more than the diameter. So, for horizontal movement (as mentioned in the appendix and end of the discussion) the interplay between forces acting on horizontal and vertical planes will be more complex. In other words, the work cannot be translated as such to horizontal locomotion.

Response:

We agree to the reviewer's comment that the explanations cannot be simply translated to horizontal rowing motion. As we introduced in lines 26-28, the horizontal locomotion has already been addressed by Hu et al¹ and others. Also, we have stated that we focus on only the vertical components of jumping on water with high inclination angle over 60° from a horizontal plane and do not intend to extend it to horizontal locomotion. When the strider propels itself across the water surface, it uses momentum transfer via vortices and capillary waves¹. In contrast, when the strider jumps high, the vertical force is mainly generated by capillary force corresponding to weight of the same volume of water as the dimples (Keller, Phys. Fluids, 10(11), 3009-3010, 1998). Therefore, the three dimensional profile of dimple shape is important to define the vertical force¹³. Because the dimples generated during high inclination jump show almost bilateral symmetry with respect to the leg, our approach would be still valid to predict vertical components of jumping. To make things clearer we modified the main text (line 348-350).

MINOR POINTS

Comment 1:

Abstract. Delete "dramatic" and "the" on line 25.

Response:

We have deleted the words.

Comment 2:

Abstract and conclusion. One may wonder (and some people do even write this very publicly) what kind of tasks such microrobots will be able to carry. Certainly NOT pollution surveillance, there are plenty of less costly ways to do so. While such hype seems to fly when writing grants, I would tone it down in publications.

Response:

As the reviewer suggested, we have toned down in suggesting possible applications. We have changed “develop biomimetic semi-aquatic microrobots” to “develop biomimetic technology” in line 25.

Comment 3:

Results. Can you give us an information about how many insects did jump with angle smaller than 60°?

Response:

We recorded total 72 jumps, of which 16 jumps were with inclination smaller than 60°. This low rate of low inclination jump is not natural but because of our experimental setup to induce vertical high jumps of water striders. No quantitative information about the frequency of jumps at different angles to horizontal in natural situation is available in any literature we know.

Comment 4:

The model was run with $M = 0.1, 0.5$ and 2 , but I wish you would do $M = 1$, as it seems to be of special interest.

Response:

Following the reviewer’s comment, we explain what value of M is the most relevant to us:

Actually, $M = 0.5$ is of the most particular interest because it reflects real water striders’ body dimensions, as mentioned in the legend of Fig. 4. Insects we observed have M varying between 0.36 and 0.76 with the average of 0.54. Therefore, we selected $M = 0.5$ as a median value of examples in Fig. 4a-f.

We speculate that the referee considered the maximum water strider’s weight the water surface can support, $mg = \sigma P$, where m is body mass, g gravitational acceleration, σ surface tension coefficient, and P perimeter of wetted part of all legs. Here, we have to distinguish $M = m/\rho l_c^2 C l_l$ used this study and $Ba = mg/(\sigma P)$. Because l_l means wetted length of a leg while P refers to perimeter of all wetted part of legs, l_l is close to

$P/8$, leading to $M \approx 8Ba$. This then implies the condition $Ba = 1$ nearly equals to $M = 8$. Under this condition, water strider cannot stably stroke, because the floating water strider's legs have already reached the maximum depth that capillary force can bear. Real water striders show much smaller Ba than 1 as reported by Hu et al.¹, therefore in this study, we do not consider M much larger than the value water striders show. We have mentioned the relation $M \approx 8Ba$ in lines 159-161 in the main text.

Comment 5:

Lines 280-282 are a repeat of previous text.

Response:

In accordance with the reviewer's comment, we have deleted the repeated sentence, "*The model confirmed that upward jumping water striders are able to maximize their jump speed*".

Comment 6:

Legend Fig. 1. Are you implying that the velocity profile in Fig. 1g, being from Koh et al., is not from the same dataset? if so, how can you use that?

Response:

In the legend of Fig. 1, we have clarified that the images and dataset of Fig.1e-h are from a single movie (lines 505). The body velocity profile of this movie in Fig. 1g has been already published by Koh et al.¹¹, so we have inserted "*Body velocity profile in (g) is the same data as that of water strider 2 in Koh et al.¹¹*" (line 511-512).

Comment 7:

Fig. 4. I am not sure Fig. 4g is a test of the model. It seems to be rather an overlay of the observed data points on the space defined by the two dimensionalized variables. Supplementary Fig. 4 is by contrast a test.

Response:

Respecting the reviewer's opinion on what is a "*test of a model*", we have changed the title of Fig. 4 to "*Theoretical and the empirical results using water striders*".

Comment 8:

Legend of Supplementary Fig. 4 (or Supplementary Fig. 5 in the revised manuscript). I do not understand the implication and rationale of the last sentence, about the overestimation of velocity. Please expand.

Response:

In accordance with the reviewer's comment, we have added sentences at the end of the legend of Supplementary Fig. 5 to support the sentence, as below.

“Overestimation of takeoff velocity in (c) may come from the delay of retraction of the water surface in the closing stage of real jump¹¹. That is, remaining dimples after the legs completely take off the water surface in Fig. 1e ($t = 25$ ms) imply that the water surface retracts slower than the legs escaping from the water surface. Therefore, dimple depth would not reflect the exact capillary force supporting the legs but exaggerate it in the closing stage.”

Comment 9:

Supplementary Figure 6 seems to be unused?

Response:

In accordance to the reviewer's suggestion, we now explain how the Supplementary Fig. 6 is complementing the remaining figures in the paper. Actually, we referred to it in the previous submitted draft, but we agree that the comment was too short. Therefore, we have inserted Supplementary Note 7 and added description of Supplementary Fig. 6, which have been referred to in lines 246-247 in the main text.

Comment 10:

A graphic showing all the distances, found in Supplementary Table 1, would be very handy.

Response:

In accordance with the reviewer's suggestion, we have inserted Supplementary Fig. 8 showing definitions of morphological measurements of water striders just below Supplementary Table 1.

Comment 11:

How do you measure Δl ?

Response:

To accommodate the reviewer's comment we explain how we measure Δl :

$\Delta l = l_l - y_i$ can be obtained from average length l_l of measured four legs of insect and the initial height of the body centre y_i from the undistorted water surface extracted from a movie. This information is available in the Supplementary Table 2.

[Reviewer 2]

MAJOR COMMENTS

Comment A:

There are a number of forces neglected in the force balance. 1) viscous or inertial drag while the leg moves along the water surface. 2) surface tension on the contact line while the leg is pulled out from the dimple. 3) Hydrostatic pressure from an air pocket formed by the leg. These forces might be small, which this reviewer is also expecting. However, it would be good to show some non-dimensional numbers or the order of magnitude calculations of these forces before the authors introduce the force (or momentum) equation.

Response:

In accordance with the ideas from the reviewer, and to introduce the force ratios before showing the equations, we have moved the explanation of force ratio from supplementary to the main text (lines 82-97) and added the ratio of energy loss due to wet adhesion when the leg becomes detached from water and kinetic energy of a water strider taking off the surface (lines 97-101). The drag on the leg moving along the water surface is not mentioned, because we only focused on the vertical components of jumping with inclination over 60° . Also, hydrostatic pressure from an air pocket formed by the leg has been already considered as buoyancy F_b in the explanation of force ratio.

Comment B:

All analyses were done in 2D projected plane. This reviewer is sure the authors already considered 3D effect, but it is not clear in the current manuscript. All measured lengths and rotation angle are in the 2D projected plane or 3D?

Response:

In accordance to the reviewer's comments, we have modified the manuscript and explain here the method of measurement of various lengths and the angle. The leg lengths including l_l , l_i , and r were measured from the images of water striders lying down flat on the ground, as described in Fig. 1c. The lengths related to the motion of water striders, such as y , Δl , l_s , and h , altogether mean the vertical length, so they can be extracted from movies. And the rotation angle of each leg in time θ_i means the angle of femur in the rotation plane of each leg with respect to the horizontal, which were estimated from the ratio of vertical length of rotating femur to real length of femur from the movie, $\theta_i = \sin^{-1}[(\text{instant vertical length of rotating femur})/(\text{length of femur})]$. We add the detailed method for the rotation angle estimation in Supplementary Note 3 and refer to it in line 139-141 in the main text.

Comment C:

Measured angular velocity has some variations in insect data as shown in Fig. 2c, which is reasonable since it is computed from animal data. So, insect data in Figure 4g should have similar fluctuations in the

measurement in angular speed of leg rotation (y-axis). The authors need to show how much variations data points in figure 4g have.

Response:

Figure 2c shows average value of angular velocity of rotating four legs extracted from a movie. We can see here, $\omega = v_{s,max}/\Delta l$ (solid line) is very close to the time average of average angular velocity $\dot{\theta}$ (dashed line), implying that using ω , a single value of each jump, as a representative angular velocity of the jump is valid. In this study, locomotion model with ω is used instead of $\dot{\theta}$, so data in Fig. 4g are not directly related to variations and fluctuations in the angular speed of leg rotation.

MINOR POINTS

Comment 1:

In Figure 1, l_v is shown instead of l_s .

Response:

Thank you for catching this. We have corrected the error.

Comment 2:

There are so many length scales for the leg (l_t , l_l , and l_w). This reviewer wonders whether the authors provide a simple schematic of the water strider to show how different lengths are defined.

Response:

Thank you for this suggestion. We have inserted Fig. 1d and modified Fig. 1c to show all the lengths the reviewer suggested.

Reviewers' Comments:

Reviewer #1 (Remarks to the Author):

The authors addressed all my comments with care, added needed information, made new graphs, added explanations and clarified the wanting points. This is a nice study.

Jérôme Casas

Reviewer #2 (Remarks to the Author):

All responses to my comments are good. One more thing I noticed while reading it once more is about the angular speed of leg rotation ω in Figure 2. As you mentioned, ω is defined as $v_{s,max}/\Delta l$. If I estimate the values from Figure 2 a & b, I got $\Delta l \sim 17$ mm and $v_{s,max} \sim 1.1$ m/s. So, I got 65 rad/s, which is higher than the solid line in Fig. 2c. Can you double check the value of ω once more?

Response to Reviewer 2

COMMENT

All responses to my comments are good. One more thing I noticed while reading it once more is about the angular speed of leg rotation ω in Figure 2. As you mentioned, ω is defined as $v_{s,max}/\Delta l$. If I estimate the values from Figure 2 a and b, I got $\Delta l \sim 17$ mm and $v_{s,max} \sim 1.1$ m/s. So, got $\omega \sim 65$ rad/s, which is higher than the solid line in Fig. 2c. Can you double check the value of ω once more?

RESPONSE

The difference of the values of ω between that in Fig. 2 (58 rad/s) and reviewer's calculation (65 rad/s) came from the differences of the values of $v_{s,max}$ and Δl used in the calculation. The values used in our calculation are such that $v_{s,max} = 1.04$ m/s and $\Delta l = 18$ mm, leading to $\omega = 58$ rad/s. To help reading the number in Fig. 2b, we added a red line corresponding to $v_{s,max}$.

Figure 2